# The Effect of Replacing Ni with Mn on the Microstructure and Properties of Al_2_O_3_-Forming Austenitic Stainless Steels: A Review

**DOI:** 10.3390/ma17010019

**Published:** 2023-12-20

**Authors:** Guoshuai Chen, Shang Du, Zhangjian Zhou

**Affiliations:** School of Materials Science and Engineering, University of Science and Technology Beijing, Xueyuan Road 30, Haidian District, Beijing 100083, China; 19910283672@163.com (G.C.); 13911755906@163.com (S.D.)

**Keywords:** austenitic stainless steel, Mn-substituted Ni, microstructure, mechanical properties, antioxidant properties

## Abstract

Al_2_O_3_-forming austenitic steel (AFA steel) is an important candidate material for advanced reactor core components due to its excellent corrosion resistance and high temperature strength. Al is a strong ferrite-forming element. Therefore, it is necessary to increase the Ni content to stabilize austenite. Ni is expensive and highly active, and so increasing the Ni content not only increases the costs but also damages the radiation resistance. Mn is a low-cost austenitic stable element. Its substitution for Ni will not only help to improve the irradiation resistance of austenitic steel, but also reduce the cost. In order to explore the feasibility of Mn-substituted Ni-stabilized austenite in AFA steel, this paper summarized the research progress of Mn-added AFA steels, whilst the research status of traditional Mn-added austenitic steels are also referred to and compared herein. The effect of the addition of Mn on the microstructure and properties of AFA steel was analyzed. The results show that Mn can promote the precipitation of the M_23_C_6_ phase and inhibit the precipitation of the B2-NiAl phase and secondary NbC phase. With the increase in Mn content, the strength of AFA steel at room temperature and high temperature decreased slightly, the room temperature elongation increased slightly, while the high temperature elongation and creep resistance decreased obviously. In addition, for austenitic steel free of Al, the addition of Mn will destroy the oxide layer of Cr_2_O_3_, which will decrease the oxidation resistance of the steel. But the preliminary study shows that Mn has little effect on the Al_2_O_3_ oxide layer. It is worth studying the effect of Mn-substituted Ni on the oxidation resistance of AFA steel. In summary, more efforts are necessary to investigate the optimal Mn content to balance the advantages and disadvantages of introducing Mn instead of Ni.

## 1. Introduction

Austenitic stainless steel is widely used in energy systems such as supercritical power plants and advanced reactors because of its excellent high-temperature mechanical properties, creep resistance, and oxidation resistance. Conventional austenitic stainless steel relies on Cr_2_O_3_ oxide film for their antioxidant properties. When applied at temperatures above 650 °C and rich in water vapor, the Cr_2_O_3_ oxide film will react with water vapor to form volatile hydroxide and lose its protective effect [1,2,3,4]. By adding the Al element to austenitic steel, it is possible to form the Al_2_O_3_ oxide-based protective film, which is the recently developed Al_2_O_3_-forming austenitic (AFA) steel [5]. Compared with the Cr_2_O_3_ oxide film, the Al_2_O_3_ oxide film is denser with higher thermodynamic stability. Therefore, the AFA steel is expected to further improve the operating parameters of an ultra-supercritical power plant and fourth-generation nuclear reactors (such as supercritical water reactors), which can significantly improve the energy conversion efficiency.

Al is a strong ferrite forming element, so it is necessary to increase the content of the Ni element to stabilize austenite. However, the Ni element is a precious metal element with scarce resources and high prices. This greatly increases the cost of AFA steel. The Mn element is also a stable element for austenite with abundant reserves. The price of the Mn element is only about 1/10 of that of the Ni element, and the price difference is still increasing [6]. In addition, Ni is a highly active element. It will undergo transmutation and produce a large amount of He in the neutron irradiation environment, resulting in material swelling, while Mn is relatively stable [7,8].

It is generally believed that the oxidation resistance of austenitic steel will be damaged by Mn substituting Ni. For example, Mn is easily diffused and has higher solubility in Cr_2_O_3_ oxide film, which can promote the conversion of Cr_2_O_3_ oxide into CrMn_1.5_O_4_ with the spinel structure, and impair the compactness of Cr_2_O_3_ oxide film [9,10,11]. However, the solubility of Mn in Al_2_O_3_ oxide film is almost zero, which does not impair the protective effect of the Al_2_O_3_ oxide film [9,10,11]. Therefore, it is possible to develop low-cost and high-performance AFA steel by substituting Mn for Ni [12,13,14,15,16,17,18]. 

The substitution of Mn for Ni in conventional austenitic steels has been studied extensively. In austenitic stainless steel welded metal, Mn can replace Mo in the σ phase, which prevents the formation of an Mo depletion zone, and thus improves the critical pitting temperature [15]. In Cr–Mn–Ni–N austenitic stainless steel, higher strength and elongation were obtained by substituting Mn for Ni [16]. In high Ni austenitic steel, the effect of 6.9 weight % Mn instead of 6 weight % Ni and the tensile properties at high temperature were improved and the cost was reduced by 10%, which made the commercial application possible [19]. However, for AFA steel, there are a few relevant research works and there is inconsistency between existing results. At present, it is generally believed that adding Mn can slightly reduce the room temperature strength and improve the room temperature elongation, but will greatly damage the antioxidant and creep properties [20,21,22]. In addition, Mn is prone to segregation, and the formation rate of the Cr_2_O_3_ oxide film is faster than that of the Al_2_O_3_ oxide film. In this case, the Mn element may promote the transformation of Cr_2_O_3_ oxide film into spinel structured CrMn_1.5_O_4_ before the formation of Al_2_O_3_ oxide film. This is the fundamental reason why adding Mn will damage the oxidation resistance of AFA steel [9]. However, this issue is also highly related to the content of Mn, and the related research work is still very limited. 

Replacing Ni with Mn has attractive prospects for AFA steel. However, the addition of Mn will also affect the mechanical properties and oxidation resistance. And currently, there is no relevant summary of the literature. This is the motivation of this work to review research works on the effect of substituting Mn for Ni on tensile properties, creep resistance, and oxidation resistance of AFA steel, and provide a reference for the study of Mn substitution for Ni in AFA steel.

## 2. Phase Composition and Microstructure

In order to analyze the effect of Mn-substituted Ni on AFA steel, steels named HC (Oak Ridge National Laboratory, Knoxville, TN, USA) series (here, HC means a high Cr content) were designed and investigated by Yamamoto et al. [20]. The specific compositions are shown in Table 1 [20]. The phases corresponding to the compositions obtained through calculation by the JMatPro software (Version 7.0, UK) are shown in Figure 1.

The results show that the content of ferrite in the HC-3 alloy is lower than that in the HC-2 alloy, indicating that the increase in Mn content has an effect on the stability of austenite. In addition, the increase in Mn content increased the content of M_23_C_6_ phase and AlN phase, decreased the content of M (C, N) phase, and caused the disappearance of the (Fe, Ni) Al phase and α-Cr phase. The addition of Mn will enlarge the austenite region. As a ferrite forming element, the solubility of Cr in austenite is relatively low. Therefore, the expansion of austenite will promote the movement of the Cr element towards grain boundaries, inducing the coarsen of Cr_23_C_6_ phase [23]. This will consume a large amount of C and the precipitation of M (C, N) phase will be restrained. 

Adding the Mn element will increase the solubility of the N element. And, the N element is very easy to combine with the Al element, forming a harmful AlN phase [9,18,24,25]. This process will consume a large amount of Al, which may be the reason for the disappearance of the NiAl phase in the HC-3 alloy compared with the HC-2 alloy. The addition of Mn can promote the precipitation of NiAl phase in ferritic steel, but not in AFA steel, because the NiAl phase and austenite are incompatible [26]. 

The precipitation of the Laves-Fe_2_Nb phase at the grain boundary would hinder Al diffusion, thereby destroying the antioxidant properties [1]. It was reported that, when Nb/C < 7.7, the precipitations of the M_23_C_6_ and NbC phases were promoted and the precipitation of the Laves-Fe_2_Nb phase can be inhibited [27]. It is necessary to conduct more in-depth investigations into the optimal Nb/C ratio in Mn-added AFA steel. With the increase in Mn content and the decrease in Ni content, a large number of ferrite phases appeared in the HC-4 alloy, which indicated that Mn-substituted Ni should fall within a suitable composition range.

In order to further confirm the effect of Mn substituting Ni on the precipitates of AFA steel, the microstructure of HC series alloys after creep fracture at 750 °C/100 MPa was analyzed, as shown in Figure 2 [20]. It can be seen that the HC-2 alloy has a lot of needle-like B2-NiAl phase both in the grain and at the grain boundary, but little M_23_C_6_ phase can be found [28]. By comparing with HC-3 alloy, it can be seen that with the increase in Mn content and the decrease in Ni content, the B2-NiAl phase in the grain boundary and grain interior obviously decreased, and the M_23_C_6_ phase content increased obviously. With the further increase in Mn content and the further decrease in Ni content, the content of B2-NiAl phase in HC-4 alloy was further decreased, and a large amount of M_23_C_6_ phase precipitated at the grain boundary and within the grain. This is consistent with the calculation results of JMatPro. The Cu-rich phase is also an important strengthening phase in austenitic steels. The size of the secondary NbC and α-Cu phases is both at the nanometer level, as shown in Figure 3. It was reported that the addition of Cu has a significant promoting effect on the precipitation of NiAl phase [29]. It is interesting to further clarify the promoting effect of Cu and the inhibitory effect of Mn on the precipitation of the NiAl phase in Mn-added AFA steel. 

In Figure 3, compared with the HC-2 alloy, the content of the secondary NbC phase in HC-4 alloy and HC-3 alloy decreased in turn, which indicated that the precipitation of secondary NbC phase in AFA steel could be inhibited by replacing Ni with Mn. All phases in Figure 3 were identified by selected area diffraction. As mentioned above, the Mn-substituted Ni promotes the precipitation of a large number of M_23_C_6_ phases at the grain boundaries, enriching C elements at the grain boundaries and thereby inhibiting the precipitation of secondary NbC phases in the grains [23]. It is worth noting that α-Cu/NbC in Figure 3 represents α-Cu precipitate phase and a secondary NbC phase. Cu-rich precipitates usually nucleate on ultra-fine carbides [30]. In conclusion, Mn-substituted Ni in AFA steel can promote the precipitation of the M_23_C_6_ phase, inhibit the precipitation of the B2-NiAl phase and secondary NbC phase, and has little effect on the α-Cu phase. The co-effect of Mn-substituted Ni, the Cu content, and of Nb/C ratio on the precipitation behavior of different phases is an interesting topic worth investigating in Mn-added AFA steels.

## 3. Tensile Properties and Creep Resistance

### 3.1. Tensile Properties

In order to analyze the effect of the Mn addition on the tensile properties of AFA steels, the content of Mn was increased on the basis of 8OC-I (12Ni-5Mn-0.2C), and the tensile data at room and high temperature were tested, as shown in Figure 4 [21]. On the basis of 10Ni-6Mn steel, the Mn content was sequentially increased to 10 wt.%, and room temperature (RT) tensile data were tested, as shown in Figure 4 [22]. The tensile strength of both 12Ni and 10Ni experimental steels slightly decreased with the increase in Mn content. The ultimate tensile strength (UTS) of 10Ni experimental steel decreased from 582 MPa slightly to 566 MPa. This indicates that, although the B2-NiAl and NbC phases precipitate decreased along with the increase in Mn content, a large number of M_23_C_6_ phase precipitated at the grain boundary and within the grain, which complemented the strengthening effect and maintained the ultimate tensile strength [31]. The UTS of 12Ni experimental steel decreased with the increase in the test temperature, which is because the thermal activation promotes the dislocation movement at high temperature, and the dislocation can more easily bypass the pinning action of precipitates by climbing and cross-sliding [32]. In Figure 4b, with the increase in Mn content, the room temperature elongation of 12Ni experimental steel decreased at first and then increased, which is consistent with the complementary strengthening effect of precipitates. And the room temperature elongation of 10Ni steel increased from 37% to 45%, which was due to the inhibition of the large-size B2-NiAl phase precipitation by adding Mn.

However, the elongation of the 12Ni experimental steel decreased from 35% to 21% with the increase in Mn content at 650 °C. This is because the decrease in B2-NiAl phase and secondary NbC phase, which are effectively strengthened phases for AFA steel [17,33,34,35,36], while the strengthening effect of the larger M_23_C_6_ phase will be weakened at a high temperature. In conclusion, adding the Mn can slightly reduce the strength of the AFA steel at room temperature and high temperature, and increase the room temperature elongation, but will greatly reduce the high temperature elongation. This is in agreement with the results of Mn substituting for Ni in traditional austenitic steels. The addition of Mn can make austenite more uniform and stable, increase the stacking fault energy in austenitic steel, promote dislocation sliding and plastic deformation, reduce the work hardening rate and hardness, and increase elongation at room temperature [14,24]. 

### 3.2. Creep Resistance

As AFA steel is mainly used in high-temperature and high-pressure environments, creep properties are very important. In order to study the effect of adding Mn on the creep properties of AFA steel, the content of Mn was increased on the basis of 8OC-I (12Ni–5Mn–0.2C). The contents of Ni, Mn, and B in HC series alloys is adjusted based on HC-1 (14Cr–5Mn–12Ni–2.5Al), as can be seen in Table 1 and Figure 5 for details [20,21]. Under the conditions of 650 °C/250 MPa and 750 °C/100 MPa, the creep life greatly decreased with the increase in Mn content. Under the conditions of 750 °C/100 MPa, the creep life decreased from 817 h to 305 h. And the creep elongation decreased from 28.9% to 8.5% at 650 °C/250 MPa.

Compared with the HC-1 alloy, the creep life of the HC-2 alloy was increased from 146 h to 484 h. The composition difference of these two steels is mainly the different content of B, which indicated that the element B was beneficial for improving the creep properties [37]. The B element enhanced the grain boundary coverage of the NiAl and Laves phases, and inhibited their coarsening, thus significantly improving the strengthening effect [38]. Compared with the HC-2 alloy, the creep life of the HC-4 and HC-3 alloys decreased in turn, which showed that the increase in the Mn substitution of Ni had an adverse effect on the creep properties.

The negative effect of a high Mn content on the creep properties of AFA steel should be due to the inhibitory effect of Mn on the precipitation of the nano-sized B2–NiAl phase and secondary NbC phase, as shown in Figure 2 and Figure 3 [36,39]. These nano-sized precipitates play a major role in the creep resistance process [33]. Furthermore, the addition of Mn promotes Cr segregation and accelerates the formation of a large sized σ phase, which is the starting point of crack propagation and damages the mechanical properties of AFA steel [15,23,36,40].

## 4. Antioxidant Properties

Excellent high-temperature oxidation resistance is also an important requirement for AFA steel [29,40,41]. Mn may strongly segregate into the external oxide layer, impairing the antioxidant properties [20]. In order to understand the segregation characteristic of Mn, the diffusion activation energy (*Q*) and pre-exponential factor (*D*_0_) of various alloying elements in austenitic steel are collected and listed in Table 2, where C, N, and B are interstitial atoms, and the others are replacement atoms [42]. The *Q* of Mn is 261.7 kJ/mol, which is much smaller than that of Ni (296.8 kJ/mol), and the *D*_0_ of Mn is only 0.16 cm^2^/s, which is about an order of magnitude smaller than that of Ni (1.09 cm^2^/s), which demonstrated that Mn has a strong segregation behavior in AFA steel. The *D*_0_ of the B element is only 0.002 cm^2^/s and its *Q* is only 87.9 kJ/mol. This indicates that the B element is also easy to segregate and enrich at the grain boundary, and thus help for grain boundary strengthening and significantly improving the creep resistance [37]. Both the *D*_0_ and *Q* of Cu are large, which is why α-Cu phase can be stable in AFA steel and contribute to excellent creep resistance [12,30].

In order to analyze the mechanism of Mn participating on oxidation, the oxidation rate constant and free energy of several typical oxides were collected as shown in Figure 6a,b. The oxidation rate constant of Al_2_O_3_ oxide layer is more than 2.5 g^2^ cm^−4^ s^−1^ lower than Cr_2_O_3_ oxide layer and the free energy is nearly 300 kJ/mol lower than Cr_2_O_3_ oxide layer, which indicates that the Al_2_O_3_ oxide film will be formed more slowly as well as be denser and more stable [2,35,39,43,44,45]. Figure 6c analyzed the mechanism of the effect of Mn addition on the oxide film of AFA steel. Mn is more prone to segregation than Al, for which there are two possible reasons. First, the strong interaction between Ni and Al makes Al fixed by the precipitated NiAl phase. Although the NiAl phase can provide Al for the formation of Al_2_O_3_ oxide film, the diffusion of Al is not as fast as that of free Mn. Secondly, the precipitates at grain boundaries, such as the Laves phase, will hinder Al diffusion along the grain boundary and retard the formation of the Al_2_O_3_ oxide film [1,46,47].

Figure 7 shows the characterization results of the oxide layer on the Mn-added AFA steel. As shown in Figure 7a, AFA steel has a composite structure on the Cr_2_O_3_ oxide layer on the outer surface and a thin Al_2_O_3_ oxide layer on the inner surface. In Figure 7b,c, the small Mn-rich island-like nodules were formed on the oxide surface of AFA steel because Mn diffused rapidly and reacted with Cr_2_O_3_ to produce spinel-like CrMn_1.5_O_4_, which is harmful to antioxidant properties. Interestingly, as can be seen from Figure 7d–f, at the site of a higher Al content on the surface oxide layer, the Mn content was 0, indicating that Mn did not affect the compactness of the Al_2_O_3_ oxide layer. The oxide layers of AFA steel from the outside to the inside were the oxide layer rich in the Fe, Cr, and Mn, the oxide layer rich in Cr_2_O_3,_ and the oxide layer rich in Al_2_O_3_, respectively [10]. This is related to oxygen activity, which is the lowest required for Al oxidation [43]. In addition, Mn and Fe have a high solubility in the Cr_2_O_3_ oxide layer and thus can easily diffuse to the surface through the Cr_2_O_3_ oxide layer. However, once the Al_2_O_3_ oxide layer formed, the flux of O_2_ diffusion inward through the oxide layer and Mn diffusion to the surface will be greatly reduced [10,43]. Therefore, it is worth investigating the relationship between Mn enrichment and the formation of alumina film.

In order to analyze the effect of Mn-substituted Ni on the oxidation resistance of AFA steels, the cyclic oxidation weight gain curves of some AFA steels with different Mn and Ni contents were calculated, as shown in Figure 8. In Figure 8a, it can be seen that the exfoliation times of 5Mn, 7Mn, and 10Mn experimental steel are 3000 h, 200 h and 700 h, respectively, which shows that 5Mn has the best antioxidation properties. In Figure 8b, it can be seen that the spalling of 14Mn–8Ni experimental steel occurred at about 3500 h, and the weight of the steel decreased obviously. However, the spalling of 9Mn–10Ni and 5Mn–12Ni experimental steel did not occur, but the weight gain rate of 9Mn–10Ni experimental steel was much higher than that of 5Mn–12Ni. The results indicated that the oxidation resistance of AFA steel will be seriously damaged if the Mn content is too high. In Figure 8c, the results showed that 0Mn sample has the slowest weight gain rate, the weight gain rate obviously increased for 0.5Mn sample. For the 1Mn sample, the exfoliation occurred after exposure for about 800 h, and the exfoliation occurred at 200 h for the 2Mn sample. Obviously, the addition of Mn decreased the antioxidation properties.

In austenitic steel, Mn will react with S to form the non-metallic inclusions of MnS. And the potential difference between MnS and matrix will result in a primary cell reaction, which will increase the pitting sensitivity of the austenitic stainless steel and impair the corrosion resistance. Therefore, the Mn content was suggested to be less than 8 weight % [48,49]. As mentioned before, the increase in Mn content will inhibit the precipitation of the NiAl phase, which can provide the Al element for the formation of the Al_2_O_3_ oxide film. Therefore, from this perspective, a higher Mn content will inhibit the formation of the Al_2_O_3_ oxide film [9,41,50]. 

In order to confirm the feasibility of Mn replacing Ni in AFA steel, the 8OC-I (5Mn-12Ni) experimental steel was compared with the commercial steels, as shown in Figure 9. From Figure 9a, it can be seen that obvious spalling occurred for 347 and Super 304H and the mass seriously decreased after oxidation for less than 500 h under the condition of 700 °C + 10 vol.% H_2_O. The mass of 709 and 310HCbN decreased first and then increased, which indicated that the oxide film may have a self-repair ability. The mass gain of 709 is slightly larger than that of 5Mn–12Ni, while the mass gain of 310HCbN is smaller than that of 5Mn–12Ni. Moreover, the mass of 5Mn-12Ni steel decreased and the oxide film peeled off after oxidation for about 3000 h, which indicated that the 5Mn–12Ni experimental steel has an oxidation resistance similar to that of commercial steel. In Figure 9b, under the condition of 650 °C + 10 vol.% H_2_O, the mass gain of 5Mn-12Ni experimental steel is very small, but the mass increase in Super 304H is obvious, and the spalling of 347 appeared at the beginning. The results show that the oxidation resistance of 5Mn–12Ni experimental steel is much better than that of 347 and Super 304H commercial steels.

## 5. Comparison with Traditional Austenitic Stainless Steel

In order to gain a more comprehensive understanding of the impact of Mn replacing Ni on austenitic stainless steel, relevant summaries were performed. Lee et al. found that in austenitic stainless steel weld metals, Mn-substituted Ni reduced the ultimate tensile strength and increased elongation (Figure 10) [15]. The increase in Ni content promoted the aggregation of Mo in σ phase [15], while Mn inhibited this effect and thus inhibited the formation of hard brittle phase-σ. This is the reason why UTS decreased and elongation increased after Mn replaced Ni. This result seems agree with that of AFA steel [21]. In Cr–Mn austenitic stainless steel, 201L is nearly 100 MPa lower than 201 [51]. This is because 201L has a higher Mn content, a coarser grain size, and thus worse mechanical properties [51]. UTS can obviously be improved through different methods, such as cold rolling, heat treatment and micro-alloying design (Figure 10) [51]. Zhang et al. found that adding V to high Mn steel can effectively improve the strength of the steel [52,53]. High Mn austenitic stainless steels are commonly used at low temperatures (77 K, 273 K), such as hydrogen storage [54,55]. This is due to the excellent crack resistance of high Mn austenitic stainless steel at low temperatures [54,56]. The potential of the application of AFA steel in a low-temperature environment is also an interesting topic. 

In order to improve the performance of cars as well as save energy and reduce emmition, light-weight has become a crucial research topic in automotive industry [57,58]. Therefore, low-density steel (LDS) has become a research hotspot [57]. Al is a light element, and it is added to 10 wt.% or so. In this case, the formation of κ-carbides ((Fe, Mn)_3_AlC) and ((Fe, Mn)_3_AlC_x_) in LDS provides high strength but relatively poor plasticity (blue circle area) [57]. The addition of precipitated phase elements such as V can form a second phase at the nanoscale [52]. This not only ensures the high strength of high manganese steel, but also improves plasticity (red circle area). Of course, a high Mn steel is not suitable for use at high temperatures due to the easy segregation of Mn elements. In AFA steel and general austenitic stainless steel, replacing Ni with Mn will reduce the strength and increase the elongation (pink and orange circle areas), as can be seen in Figure 10.

In terms of antioxidant properties, it was found that the weight gain rate of 3Mn was higher than that of 1Mn, as seen in Figure 11 [59]. Lu et al. compared the samples of 0.65-Mn, 2-Mn, and 8-Mn, and found that the oxidation weight increased with the increase in Mn content [60]. The mass gain of this process is mainly a Mn-rich layer (Mn_2_O_3_, Mn_3_O_4_ and FeMnO_3_) [60,61]. Su et al. added 9 wt.% Mn to austenitic steel, and after oxidation for 100 h, the mass greatly increased [62]. The obvious effect of the addition of Al on the antioxidant properties is highlighted here. Zhao et al. suggest that the Mn content in AFA steel controlled in the range of 6–8 wt.% showed better oxidation resistance (800 °C, in air) [63,64] compared with the data from Brady et al., in which the content of Mn is 2 wt.% [21]. This indicated that, generally, the weight gain after high-temperature oxidation increased along with the increase in Mn content, but the story in AFA steel is still not clear, and it seems that, in the high-temperature environment absence of water vapor, the Mn content can be appropriately high (Figure 11).

Figure 12 collected and summarized the relationship between the content of Mn and creep performance. It is obvious that Mn-substituted Ni will seriously damage the creep resistance. At high temperature, the Mn diffuses rapidly to the surface, so the effect of stabilizing austenite near the surface will be weakened, which will influence the creep resistance. Therefore, the amount of Mn added to austenitic steel used at high temperature must be very limited. Fe–25Ni–0Mn has a very high creep life, as shown in Figure 12. The reason is that, in addition to Mn-free, the addition of pre-strain and Ti promotes the formation of a large number of very fine carbides (TiC/secondary NbC) which is beneficial for the improvement of creep properties [65,66]. From this point of view, it is possible to make up the loss of creep properties caused by adding Mn by adjusting the treatment process and micro-alloying the composition design. 

## 6. Conclusions

Replacing Ni with Mn has attracted widespread attention and become an important research topic for the development of AFA due to the advantage of cost reduction and the improvement of irradiation resistance. However, the addition of Mn will also influence the microstructure and properties of AFA steels. The related research works have been reviewed and discussed. The conclusions were as follows:(1)Increasing the Mn content and decreasing the Ni content in AFA steel will suppress the precipitation of the B2–NiAl phase and NbC phase, but increase the M_23_C_6_ phase while the nano-scale α-Cu phase is not affected. The room-temperature and high-temperature ultimate tensile strength slightly decreased, the room-temperature elongation slightly increased, and high-temperature elongation significantly decreased with the increase in Mn content. Further research is needed to determine the optimal amount of substitution content of Mn for Ni.(2)The substitution of Mn for Ni seriously damages the creep properties. Mn promotes σ phase precipitation which can easily initiate crack propagation. Therefore, it is worth studying how can the σ phase precipitation be restrained through composition design and mechanical heat treatment for the Mn-added AFA steel.(3)Mn is easy to segregate and has high solubility in the oxide layer of Cr_2_O_3_, which promotes the conversion of Cr_2_O_3_ oxide into the spinel CrMn_1.5_O_4_ and damages the antioxidant properties. But the solubility of Mn in the oxide layer of Al_2_O_3_ is nearly zero. Determining how the segregation of Mn can be restrained, how the formation rate of Al_2_O_3_ oxide film can be accelerated, and how the loss of Mn by anti-oxidation elements such as Si can be compensated are the goals of future research.(4)In AFA steel and general austenitic stainless steel, the effect of Mn replacing Ni basically shows the same trend. Replacing Ni with Mn will reduce UTS and improve plasticity. And, it can damage the oxidation resistance and creep resistance. More efforts are necessary to investigate the effect of thermal mechanical treatment and composition design on property improvement to compensate for Mn-substituted Ni.

## Figures and Tables

**Figure 1 materials-17-00019-f001:**
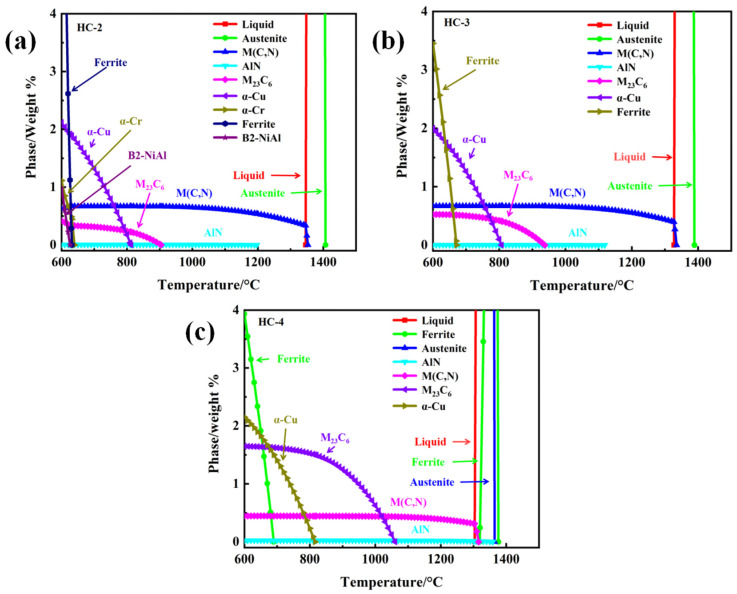
JMatPro calculation results of the HC series alloys: (**a**) HC-2; (**b**) HC-3; and (**c**) HC-4.

**Figure 2 materials-17-00019-f002:**
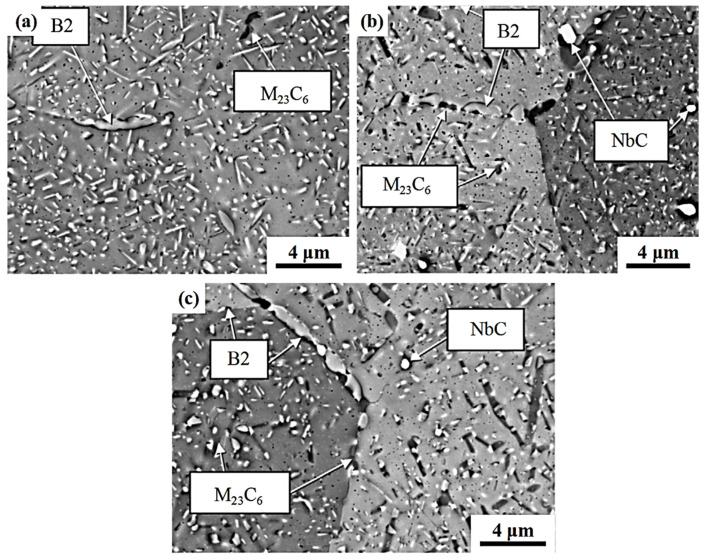
Backscatter image of the microstructure of the HC series alloys after creep fracture at 750 °C/100 MPa: (**a**) HC-2; (**b**) HC-3; and (**c**) HC-4 [20].

**Figure 3 materials-17-00019-f003:**
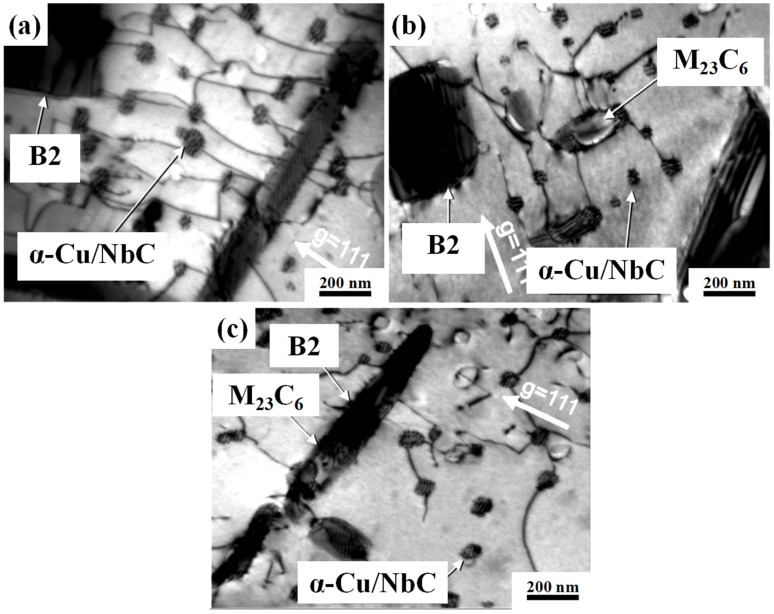
The bright field-TEM image of HC series alloys after creep fracture at 750 °C/100 MPa: (**a**) HC-2; (**b**) HC-3; and (**c**) HC-4 [20].

**Figure 4 materials-17-00019-f004:**
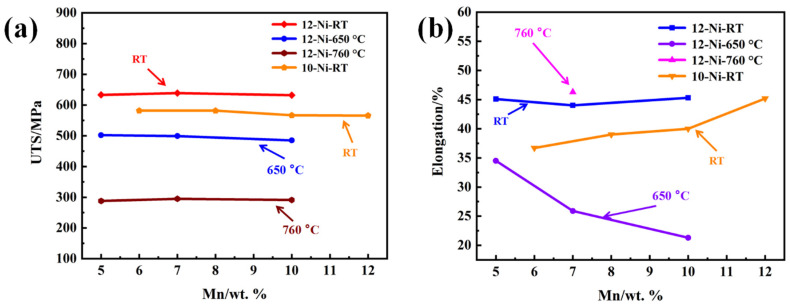
The relationship between Mn content and tensile properties of AFA steels: (**a**) UTS; (**b**) elongation (RT—room temperature) [21,22].

**Figure 5 materials-17-00019-f005:**
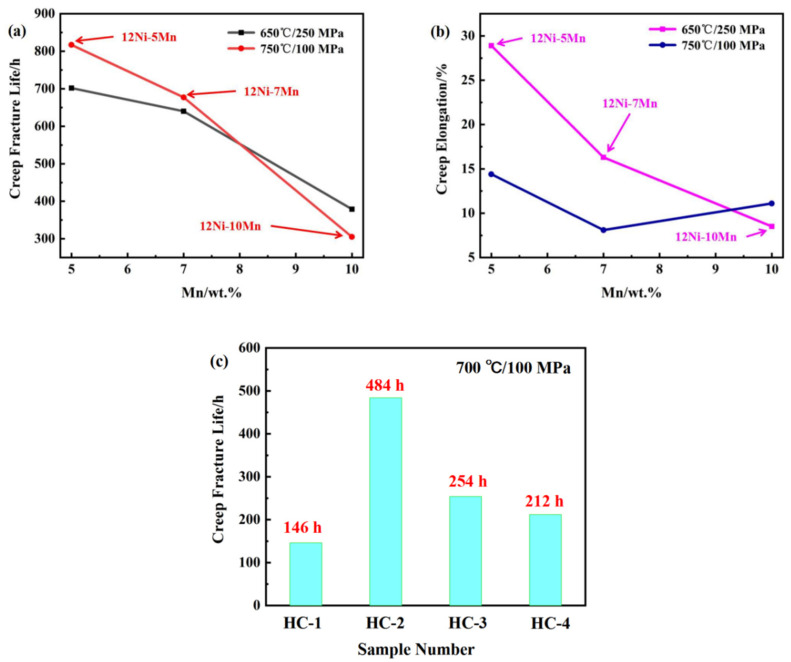
AFA steel creep resistance changes with Mn content: (**a**) creep fracture life; (**b**) creep elongation; and (**c**) HC series alloys [20,21].

**Figure 6 materials-17-00019-f006:**
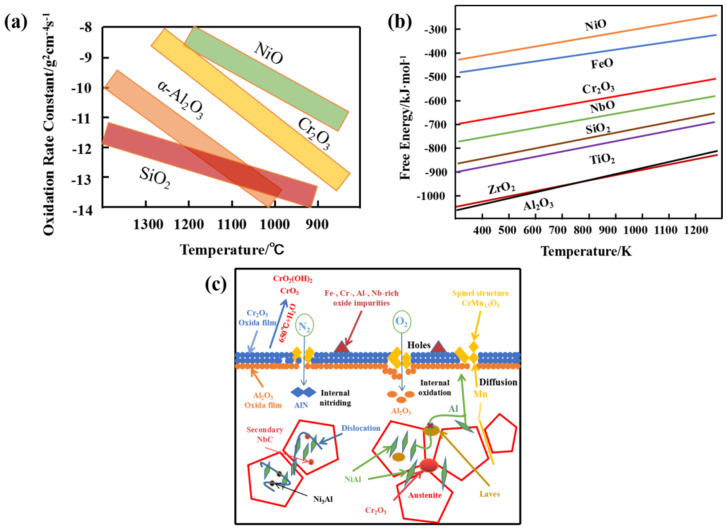
The properties of common oxides changed with temperature and the effect of Mn: (**a**) oxidation rate constant; (**b**) free energy [9]; and (**c**) the mechanism diagram of Mn participating in the formation of oxide film.

**Figure 7 materials-17-00019-f007:**
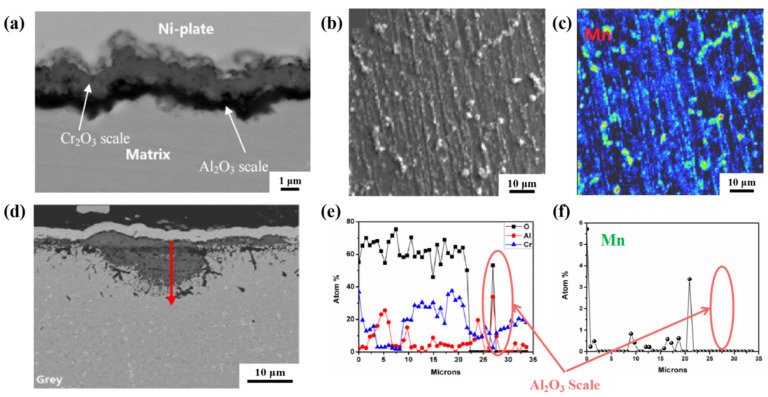
Characteristics of the oxide layer on the AFA steel surface: (**a**) a cross-section of the oxide layer; (**b**) the outer surface of the oxide layer; (**c**) surface scanning electron probe microanalysis of oxide surface; (**d**) section topography of oxide layer; (**e**) the distribution of O, Al, and Cr in the oxide layer by scanning along the red arrow line; and (**f**) the distribution of Mn in the oxide cross-section scanned along the red arrow line (The red arrow is to indicate the position where Al_2_O_3_ appears, and the Mn element is 0.) [9].

**Figure 8 materials-17-00019-f008:**
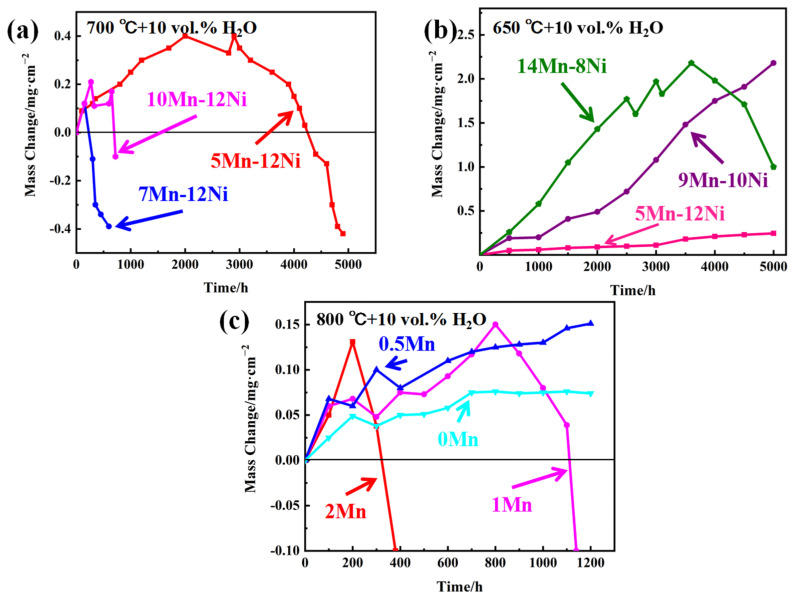
Weight gain curves of several AFA steels with Mn content: (**a**) 700 °C + 10 vol.% H_2_O; (**b**) 650 °C + 10 vol.% H_2_O; and (**c**) 800 °C + 10 vol.% H_2_O [9,20,21].

**Figure 9 materials-17-00019-f009:**
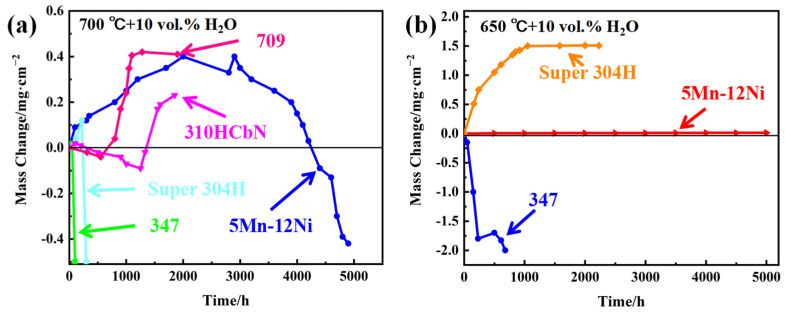
Comparison of oxidation gain curves of 8OC-I (12Ni-5Mn-0.2C) steels and commercial steels under different oxidation conditions: (**a**) 700 °C + 10 vol.% H_2_O, (**b**) 650 °C + 10 vol.% H_2_O [21].

**Figure 10 materials-17-00019-f010:**
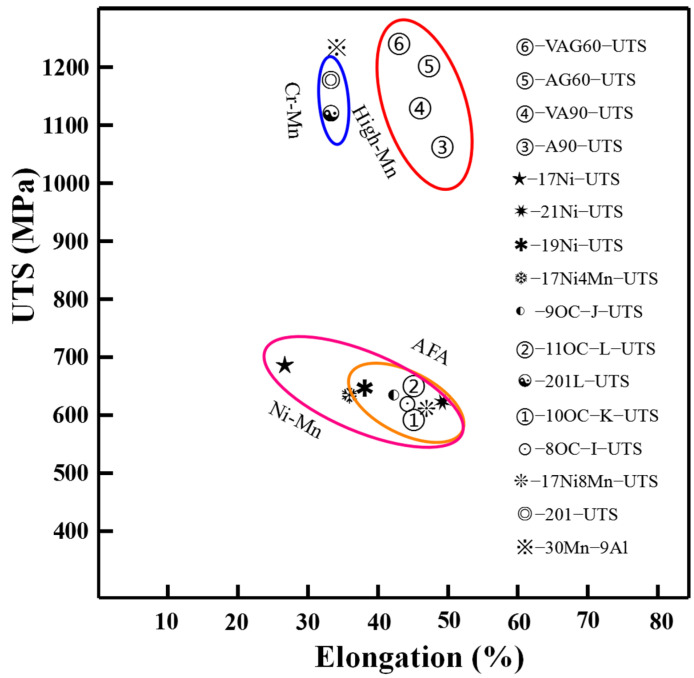
Summary of mechanical properties in Mn containing austenitic steel [15,21,51,52,57].

**Figure 11 materials-17-00019-f011:**
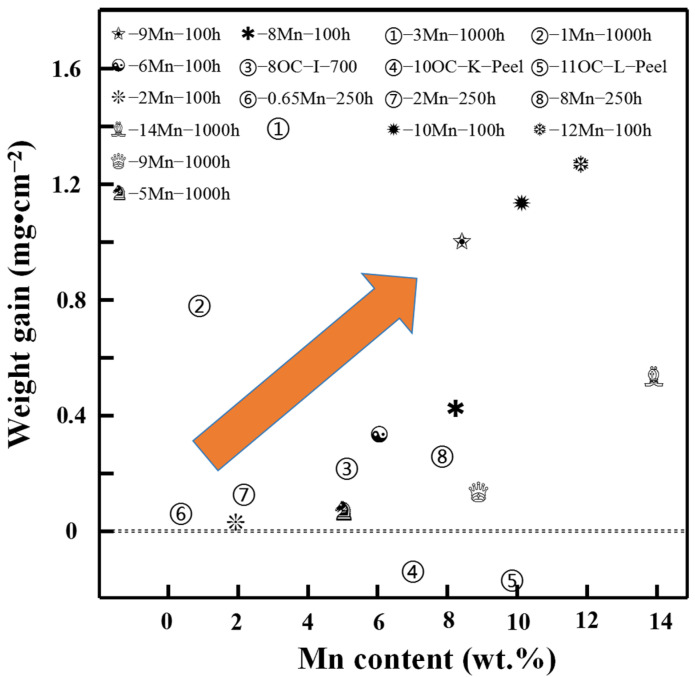
The trend of the oxidation weight gain in Mn containing austenitic stainless steel with respect to Mn content [20,21,58,59,60,61].

**Figure 12 materials-17-00019-f012:**
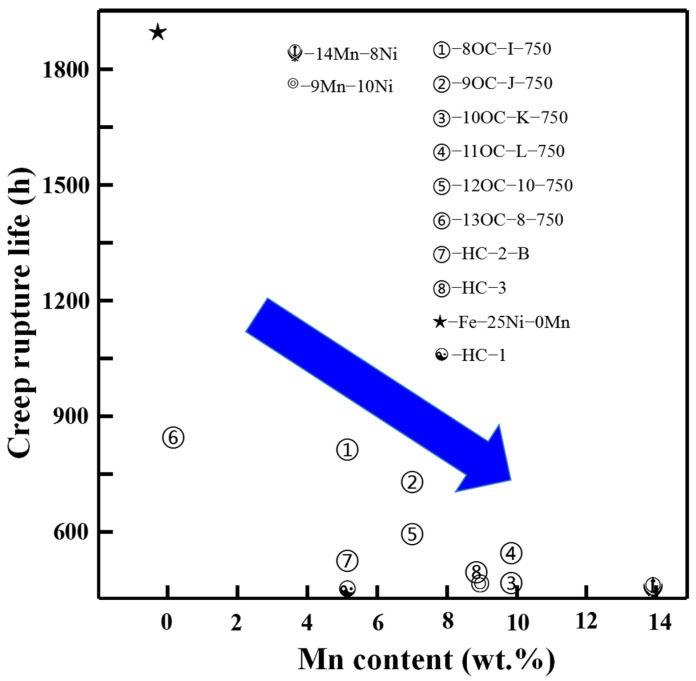
The trend of creep fracture life with the Mn content in austenitic stainless steel containing Mn [20,21,62].

**Table 1 materials-17-00019-t001:** HC series AFA steel composition, wt.% [20].

Sample Name	Fe	Cr	Mn	Ni	Cu	Al	Nb	C	B	N
HC-1	63.3	14	4.7	12	2.9	2.4	0.6	0.1	0.001	0.001
HC-2	63.1	14	4.7	12	3	2.5	0.6	0.089	0.007	0.002
HC-3	60.5	14	9.3	10.1	2.9	2.4	0.6	0.1	0.013	0.001
HC-4	58.2	14.2	13.6	8.2	3	2.4	0.4	0.14	0.015	0.008

**Table 2 materials-17-00019-t002:** Diffusion activation energy *Q* and pre-exponential factor *D*_0_ of alloying elements in austenite [42].

Element Name	*D*_0_/cm^2^∙s^−1^	*Q*/kJ∙mol^−1^	Element Name	*D*_0_/cm^2^∙s^−1^	*Q*/kJ∙mol^−1^
C	0.738	159	Cr	4.08	286.8
N	0.043	123	Cu	4.16	306.2
B	0.002	87.9	Mn	0.16	261.7
Fe	1.05	283.9	Ni	1.09	296.8
Co	1.25	305.2	V	800.01	330.3
W	1000	376.8	Mo	0.036	239.8

## Data Availability

Data will be available on request.

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
