# Peer review of "The Effect of Replacing Ni with Mn on the Microstructure and Properties of Al2O3-Forming Austenitic Stainless Steels: A Review"

_materials, 2023, doi:10.3390/ma17010019_

Round 1
Reviewer 1 Report
Comments and Suggestions for Authors
1. Is it correct name "Cu particles" in matrix? Maybe Cu-rich particles better describes this?
2. What is Cu/NbC phase? Cu does not create carbides. Acccording to literature (E. g. DOI: 10.1017/S1431927614013221) Cu rich precipitates have nucleated on carbides. Have Authors got some data about type of crystalic structure of this Cu/NbC?
3. Fig. 3 has showed types of precipitates, but without dirfraction or EDS spectra. How they were identified? Please add this information.
Comments on the Quality of English Language
Some sentences are too long.
Author Response
Response to Editor and Reviewer's Comments
Dear Editor and Reviewers,
We would like to express our great appreciation to you for your constructive comments and suggestions on our manuscript entitled “The effect of replacing Ni with Mn on the microstructure and properties of Al2O3-forming austenitic stainless steels: a review”. The manuscript has been carefully revised according to the editor and reviewers' comments. Following is the detailed response to editor and reviewers’ Comments:
Reviewer#1: 1. Is it correct name "Cu particles" in matrix? Maybe Cu-rich particles better describes this?
Response: Thanks for reviewer’s kindly reminding. We agree that it is better to use Cr-rich particles instead of Cu particles. According to the content in reference [20] (DOI: 10.1016/j.msea.2009.06.043), the specific Cu here is in the form of α-Cu phase. We have re-written the related sentences in the revised manuscript. Furthermore, the manuscript has been checked and revised carefully to improve the writing, discussion and organization.
- What is Cu/NbC phase? Cu does not create carbides. Acccording to literature (E. g. DOI: 10.1017/S1431927614013221) Cu rich precipitates have nucleated on carbides. Have Authors got some data about type of crystalic structure of this Cu/NbC?
Response: Thanks for reviewer’s kindly reminding. According to reference [20] 10.1017/j.msea.2009.06.043), these nanoscale precipitates marked as Cu/NbC should mainly be α-Cu precipitated phase and secondary NbC phase. Therefore, we think that the authors using Cu/NbC may means α-Cu precipitate phase or secondary NbC precipitate phase. This is consistent with the results calculated in Figure 1, as well as the literature (DOI: 10.1017/S1431927614013221) you referred. NbC phase and Cu-rich precipitate phase are distributed adjacent to each other. We have revised the related sentence to make it more clearly. The literature you mentioned was also added.
- Fig. 3 has showed types of precipitates, but without dirfraction or EDS spectra. How they were identified? Please add this information.
Response: Thanks for reviewer’s kindly reminding. According to the content of reference [20] (DOI: 10.1017/j.msea.2009.06.043), these precipitates were determined by TEM-XEDS, especially by SADE (selected area diffraction). We have added the related information in the revised manuscript.
- Some sentences are too long.
Response: Thanks for reviewer’s kind suggestion. The long obscure sentences have been rewritten for better expression and understandable in the revised manuscript.
Guoshuai Chen
E-mail: 19910283672@163.com
Corresponding author:
Name: Zhangjian Zhou
E-mail: zhouzhj@mater.ustb.edu.cn
Address: School of Materials Science and Engineering,
University of Science and Technology Beijing, Beijing 100083, China
Thank you again for your valuable comments and suggestions!
Reviewer 2 Report
Comments and Suggestions for Authors
The review manuscript describes the effect of replacing Ni with Mn on structural, mechanical, and antioxidant properties. In the entire review manuscript, the authors are based specifically on the results from 4 sources, which are listed in the list of references as: 9; 20; 21; and 22. The given issue would deserve deeper processing and analysis of results from several sources.
The manuscript is well organized overall, and the topic is related to the interest of this journal. However, before this manuscript becomes acceptable, the authors are required to address the following comments:
There is a line numbering error in the submitted manuscript, and therefore it is difficult to indicate the exact location of the review.
- Most of the statements in the manuscript consist of long and complex sentences, which are hard for the readers to understand.
- Please replace the term property with properties.
- In chapter 2, phase composition and microstructure, explain the acronym "HC series".
- In Table 1, the conception of HC-1 and HC-2 is almost the same. What is the point of stating both of these conceptions?
- Figure 3 is a BF-TEM image or bright field TEM image.
- Do not enter decimal numbers in the values of mechanical properties: 3.1 Tensile properties: "experimental steel decreased from 581.75 MPa to 565.7 MPa".
- Austenitic stainless steels are also used at cryogenic temperatures. It would be appropriate to expand the section „3.1 Tensile properties“ to include the properties of these steels in cryogenic conditions (77K; 4.2K).
- In Figure 4, the authors show the strength values. The strength is a set of the YS (yield strength) and the UTS (ultimate tensile strength). Please indicate whether it is YS or UTS.
- "Author Contributions" are missing from the manuscript. Please complete.
- A review article should be based on a large amount of used literature to guarantee its quality. I recommend expanding the references used.
A review article should provide a broader view of the issue. It would be appropriate to expand the presented review manuscript with other concepts and compare it with several authors. In the case of the properties of these steels, the author of the manuscript is based on only one source, and there is no comparison with other studies. I strongly recommend expanding the chapter on properties to include other studies.
The paper should be proofread and edited carefully to ensure clarity, accuracy, and consistency in language and formatting.
Based on the abovementioned comments, this manuscript is recommended for major revision. A revised version is required.

Author Response
Response to Editor and Reviewer's Comments
Dear Editor and Reviewers,
We would like to express our great appreciation to you for your constructive comments and suggestions on our manuscript entitled “The effect of replacing Ni with Mn on the microstructure and properties of Al2O3-forming austenitic stainless steels: a review”. The manuscript has been carefully revised according to the editor and reviewers' comments. Following is the detailed response to editor and reviewers’ Comments:
Reviewer#2: The review manuscript describes the effect of replacing Ni with Mn on structural, mechanical, and antioxidant properties. In the entire review manuscript, the authors are based specifically on the results from 4 sources, which are listed in the list of references as: 9; 20; 21; and 22. The given issue would deserve deeper processing and analysis of results from several sources. The manuscript is well organized overall, and the topic is related to the interest of this journal. However, before this manuscript becomes acceptable, the authors are required to address the following comments: There is a line numbering error in the submitted manuscript, and therefore it is difficult to indicate the exact location of the review.
- Most of the statements in the manuscript consist of long and complex sentences, which are hard for the readers to understand.
Response: Thanks very much for reviewer’s kind comments and suggestion. The long and complex sentences have been rewritten in the revised manuscript for better expression and easier understanding.
- Please replace the term property with property with properties.
Response: Thanks for reviewer’s kindly reminding. The term has been revised accordingly. Furthermore, the manuscript has been checked and revised carefully to improve the writing, discussion and organization.
- In chapter 2, phase composition and microstructure, explain the acronym “HC series”.
Response: Thanks for reviewer’s kindly reminding. Explanation of HC series has been added in the revised manuscript. According to the content of reference [20] (DOI: 10.1016/j.msea.2009.06.043), the author divided the samples into three types based on their composition: HA (10Cr-2.5Al), HB (12Cr-3Al), and HC (14Cr-2.5Al). HA should represent high Al content. HC refers to having a higher Cr content compared to HA. HB should represent a higher content of both Cr and Al compared to HA. This article only uses HC series alloys to illustrate the effect of Mn replacing Ni.
- In Table 1, the conception of HC-1 and HC-2 is almost the same. What is the point of stating both of these conceptions?
Response: Thanks for reviewer’s kindly reminding. The biggest difference between HC-1 and HC-2 is the difference in B content, which is 0.001 wt.% and 0.007 wt.% in HC-1 and HC-2, respectively. The creep fracture lives of HC-1 and HC-2 under 700 °C/100 MPa conditions are 146 hours and 484 hours, respectively. It is considered that the addition of B resulting the improvement of creep performance of HC-1, the related information was added in the part of 3.2 in the revised manuscript.
- Figure 3 is a BF-TEM image or bright fieldTEM image.
Response: Thanks for reviewer’s kindly reminding. Figure 3 is bright field TEM image. We have revised it.
- Do not enter decimal numbers in the values of mechanical properties: 3.1 Tensile properties : “experimental steel decreaesd from 581.75 MPa to 565.7 MPa”.
Response: Thanks for reviewer’s kindly reminding. We have made the modifications based on your suggestions.
- Austenitic stainless steels are also used at cryogenic temperatures. It would be appropriate to expand the section “1 Tensile properties”to include the properties of these steels in cryogenic conditions (77K; 4.2K).
Response: Thanks for reviewer’s kindly reminding. According to your suggestion, we have added descriptions of austenitic stainless steel for low-temperature use in the newly added Chapter 5. Austenitic steels with high Mn content is not suitable for high temperature application as Mn is prone to segregation at high temperatures, while for low temperature application, it is possible to added a higher content of Mn in austenitic stainless steel. Thank you for your careful review.
- In Figure 4, the authors show the strength values. The strength is a set of the YS (yield strength) and the UTS (ultimate tensile strength). Please indicate whether it is YS or UTS.
Response: Thanks for reviewer’s kindly reminding. Ultimate tensile strength (UTS) and yield strength (YS) are completely different concepts. In Figure 4, it is mainly the ultimate tensile strength (UTS). We have modified it in the revised manuscript.
- “Author Contributions”are missing from the manuscript. Please complete.
Response: Thanks for reviewer’s kindly reminding. We have added the author contributions section.
- A review article should be based on a large amount of used literature to guarantee its quality. I recommend expanding the references used.
Response: Thanks for reviewer’s kindly reminding. Replacing Ni with Mn in AFA steel is a new research topic, resulting in a quite limited relevant literature at present. Based on your suggestion, we have expanded the content of this manuscript. More references were added to assist in explaining the impact of Mn replacing Ni on the performance of AFA steels. Some related reference on the role of Mn replacing Ni in traditional austenitic stainless steel were also added to expand the references used.
- A review article should provide a broader view of the issue. It would be appropriate to expand the presented review manuscript with other concepts and compare it with several authors. In the case of the properties of these steels, the author of the manuscript is based on only one source, and there is no comparison with other studies. I strongly recommend expanding the chapter on properties to include other studies. The paper should be proofread and edited carefully to ensure clarity, accuracy, and consistency in language and formatting.Based on the abovementioned comments, this manuscript is recommended for major revision. A revised version is required.
Response: Thanks for reviewer’s comments. We fully agree them. The manuscript was carefully checked and revised to improve language expression and clarity, accuracy and consistency. We have added a new section 5 in the revised manuscript, which mainly introduced the comparative results of the effects of Mn replacing Ni in AFA steel and other traditional austenitic steels. A large number of references were cited and compared in detail with the achievements of others. Thank you very much for your constructive suggestions
Guoshuai Chen
E-mail: 19910283672@163.com
Corresponding author:
Name: Zhangjian Zhou
E-mail: zhouzhj@mater.ustb.edu.cn
Address: School of Materials Science and Engineering,
University of Science and Technology Beijing, Beijing 100083, China
Thank you again for your valuable comments and suggestions!
Reviewer 3 Report
Comments and Suggestions for Authors
This paper reviews Mn effect on the properties of alumina forming austenitic stainless steels. The paper reviews a very specific topic and basically reviews mostly just one paper, i.e. Evaluation of Mn substitution for Ni in alumina-forming austenitic stainless steels, Yamamoto et al., Materials Science and Engineering A 524 (2009) 176 (reference number 20). All the important information that had been considered in this review paper has been already addressed in the above research paper. The conclusions drawn by this review paper are also already addressed by the above paper. For this reason, the reviewer thinks that the potential readers of this paper cannot get any better insight from this review paper when comparing that with the case of reading the research paper of reference 20. Therefore, the reviewer cannot recommend this paper to be published in the current form.
In the reviewer's opinion, the topic of the paper should be greatly extended to the general Mn replacement effect to Ni for the austenitic stainless steel, for the publication of the review paper.
Author Response
Response to Editor and Reviewer's Comments
Dear Editor and Reviewers,
We would like to express our great appreciation to you for your constructive comments and suggestions on our manuscript entitled “The effect of replacing Ni with Mn on the microstructure and properties of Al2O3-forming austenitic stainless steels: a review”. The manuscript has been carefully revised according to the editor and reviewers' comments. Following is the detailed response to editor and reviewers’ Comments:
Reviewer#3: This paper reviews Mn effect on the properties of alumina forming austenitic stainless steels. The paper reviews a very specific topic and basically reviews mostly just one paper, i.e. Evaluation of Mn substitution for Ni in alumina-forming austenitic stainless steels, Yamamoto et al., Materials Science and Engineering A 524 (2009) 176 (reference number 20). All the important information that had been considered in this review paper has been already addressed in the above research paper. The conclusions drawn by this review paper are also already addressed by the above paper. For this reason, the reviewer thinks that the potential readers of this paper cannot get any better insight from this review paper when comparing that with the case of reading the research paper of reference 20. Therefore, the reviewer cannot recommend this paper to be published in the current form. In the reviewer's opinion, the topic of the paper should be greatly extended to the general Mn replacement effect to Ni for the austenitic stainless steel, for the publication of the review paper.
Response: Thank you very much for your valuable comments and suggestions. Replacing Ni with Mn in AFA steel is a new research topic, the related literature is quite limited at present, it is true that the main content of this review comes from references [20] (DOI: 10.1016/j.msea.2009.06.043), [21] (DOI: 10.1016/j.msea.2013.10.014), and [22] (DOI: 10.1038/s41598-023-32968-9). We think Mn substation Ni in AFA steel is very interesting and some issues are highly worthy to in-depth research in the near future. This is the motivation of this manuscript. Apart from the main references, we also cited many other related literatures and added our own works, such as the thermal dynamic calculation to strengthen the discussion and forecast the future development in this topic. Furthermore, we have added a new section in the revised manuscript based on your suggestion to expand the review of Mn replacing Ni to the traditional austenitic steel. The comparison between traditional austenitic steel and AFA steel is significant to better understand the reported results and consider the future development directions.
Guoshuai Chen
E-mail: 19910283672@163.com
Corresponding author:
Name: Zhangjian Zhou
E-mail: zhouzhj@mater.ustb.edu.cn
Address: School of Materials Science and Engineering,
University of Science and Technology Beijing, Beijing 100083, China
Thank you again for your valuable comments and suggestions!
Round 2
Reviewer 2 Report
Comments and Suggestions for Authors
Dear authors, Thank you for adding my comments. I confirm that all comments proposed by me have been incorporated. The manuscript has been extensively revised and edited. I recommend accepting the manuscript.
Reviewer 3 Report
Comments and Suggestions for Authors
The paper is improved greatly. I recommend this paper to be accepted.